# MVP-BERT: REDESIGNING VOCABULARIES FOR CHINESE BERT AND MULTI-VOCAB PRETRAINING

## ABSTRACT

Despite the development of pre-trained language models (PLMs) significantly raise the performances of various Chinese natural language processing (NLP) tasks, the vocabulary for these Chinese PLMs remain to be the one provided by Google Chinese Bert Devlin et al. (2018), which is based on Chinese characters. Second, the masked language model pre-training is based on a single vocabulary, which limits its downstream task performances. In this work, we first propose a novel method, *seg_tok*, to form the vocabulary of Chinese BERT, with the help of Chinese word segmentation (CWS) and subword tokenization. Then we propose three versions of multi-vocabulary pretraining (MVP) to improve the models expressiveness. Experiments show that: (a) compared with char based vocabulary, *seg_tok* does not only improves the performances of Chinese PLMs on sentence level tasks, it can also improve efficiency; (b) MVP improves PLMs' downstream performance, especially it can improve *seg_tok*'s performances on sequence labeling tasks.

## 1 INTRODUCTION

The pretrained language models (PLMs) including BERT Devlin et al. (2018) and its variants Yang et al. (2019); Liu et al. (2019b) have been proven beneficial for many natural language processing (NLP) tasks, such as text classification, question answering Rajpurkar et al. (2018) and natural language inference (NLI) Bowman et al. (2015), on English, Chinese and many other languages. Despite they brings amazing improvements for Chinese NLP tasks, most of the Chinese PLMs still use the vocabulary (vocab) provided by Google Chinese Bert Devlin et al. (2018). Google Chinese Bert is a character (char) based model, since it splits the Chinese characters with blank spaces. In the pre-BERT era, a part of the literature on Chinese natural language processing (NLP) first do Chinese word segmentation (CWS) to divide text into sequences of words, and use a word based vocab in NLP models Xu et al. (2015); Zou et al. (2013). There are a lot of arguments on which vocab a Chinese NLP model should adopt.

The advantages of char based models are clear. First, Char based vocab is smaller, thus reducing the model size. Second, it does not rely on CWS, thus avoiding word segmentation error, which can directly result in performance gain in span based tasks such as named entity recognition (NER). Third, char-based models are less vulnerable to data sparsity or the presence of out-of-vocab (OOV) words, and thus less prone to over-fitting (Li et al. (2019)). However, word based model has its advantages. First, it will result in shorter sequences than char based counterparties, thus are faster. Second, words are less ambiguous, thus may be helpful for models to learn the semantic meanings of words. Third, with a word based model, exposure biases may be reduced in text generation tasks (Zhao et al. (2013)). Another branch of literature try to strike a balance between the two by combining word based embedding with character based embedding Yin et al. (2016); Dong et al. (2016).

In this article, we try to strike a balance between the char based model and word based model and provides alternative approaches for building a vocab for Chinese PLMs. In this article, there are three approaches to build a vocab for Chinese PLMs: (1) following Devlin et al. (2018), separate the Chinese characters with white spaces, and then learn a sub-word tokenizer (denote as *char*); (2) first segment the sentences with a CWS toolkit like jieba[1], and then learn a sub-word tokenizer (denoted

---

[1]https://github.com/fxsjy/jieba

as *seg_tok*); (3) do CWS and kept the high-frequency words as tokens and low-frequency words will be tokenized by *seg_tok* (denoted as *seg*). See Figure 1 for their workflow of processing an input sentence. Note that the first one is essentially the same with the vocab of Google Chinese BERT.

Inspired by the previous work that incorporate multiple vocabularies (vocabs) or combine multiple vocabs in an natural wayYin et al. (2016); Dong et al. (2016), we also investigate a series of strategies, which we will call Multi-Vocab Pretraining (MVP) strategies. The first version of MVP is to incorporate a hierarchical structure to combine the char based vocab and word based vocab. From the viewpoint of model forward pass, the embeddings of Chinese characters are aggregated to form the vector representations of multi-gram words or tokens, which then are fed into transformer encoders, and then the word based vocab will be used in masked language model (MLM) training. We will denote this version of MVP as $MVP_{hier}$. Note that in $MVP_{hier}$, the char based vocab is built by splitting the Chinese words in the word based vocab into Chinese chars, and non-Chinese tokens are kept the same. We will denote this strategy as $MVP_{hier}(V)$, where $V$ is a word based vocab.

The second version of MVP (denoted as $MVP_{pair}$) is to employ a pair of vocabs in MLM. Due to limited resources, in this article we only consider the pair between *seg_tok* and *char*. $MVP_{pair}$ is depicted in Figure 2(c). In $MVP_{pair}$, a sentence (or a concatenation of multiple sentences in pretraining), is processed and tokenized both in *seg_tok* and *char*, and the two sentences are encoded by two parameter-sharing transformer encoders. Whole word masking Cui et al. (2019b) is applied for pretraining. For example, the word "篮球" (basketball) is masked. The left encoder, which is with *seg_tok*, has to predict the single masked token is "篮球", and the right encoder has to predict "篮" and "球" for two masked tokens. MLM loss from both sides will be added with weights. With $MVP_{pair}$, parameter sharing enables the single vocab model to absorb information from the other vocab, thus enhancing its expressiveness. Note that after pre-training, one can either keep one of the encoder or both encoders for downstream finetuning. We will denote this strategy as $MVP_{pair}(V_1, V_2, i)$, where $V_1$ and $V_2$ are two different vocabs, $i = s$ means only the encoder with $V_1$ is kept for finetuning (single vocab model), and $i = e$ means both encoders are kept (ensemble model).

The third version of MVP (denoted as $MVP_{obj}$) is depicted in Figure 2(b). In $MVP_{obj}$, the sentence is encoded only once with a fine-grained vocab, and MLM task with that vocab is conducted. As in the figure, he word "喜欢" (like) is masked, and under the vocab of *char*, the PLM has to predict "喜" and "欢" for the two masked tokens. As additional training objective, we will employ a more coarse-grained vocab like *seg_tok* and ask the model to use the starting token ("喜")'s representation to predict the original word under *seg_tok*. We will denote this strategy as $MVP_{pair}(V_1, V_2)$, where $V_1$ and $V_2$ are a pair of vocabs and $V_1$ is the more fine-grained one.

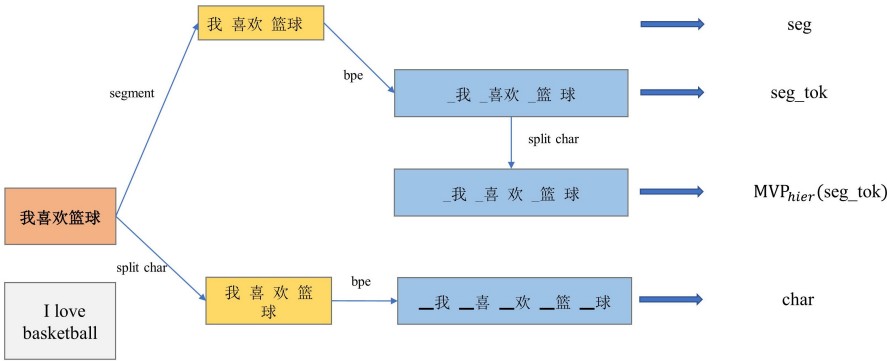

Figure 1: An illustration of how to process input sentence into tokens under different methods we define.

Extensive experiments and ablation studies are conducted. We select BPE implemented by sentencepiece[2] as the sub-word model, and Albert **?** (base model) as our PLM. Pre-training is done on Chinese Wikipedia corpus[3], which is also the corpus on which we build the different vocabs. After

---

[2]https://github.com/google/sentencepiece
[3]https://dumps.wikimedia.org/zhwiki/latest/

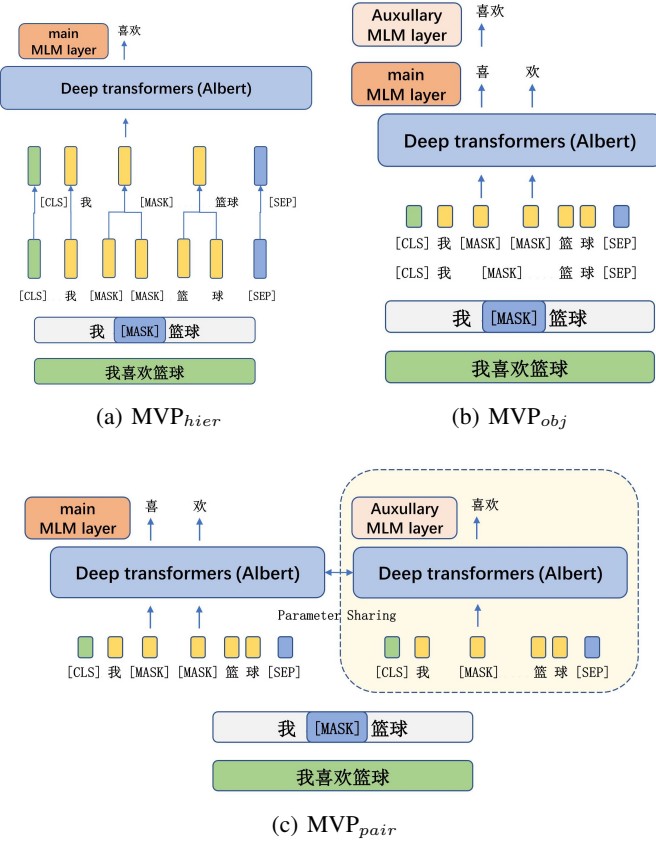

Figure 2: The architectures for the three versions of MVP strategies.

pretraining, the three vocab building methods and three MVP strategies are compared on a series of Chinese benchmark datasets, 4 of which are sentence classification tasks, 3 are sequence labeling tasks. The experimental results demonstrate the following take-aways: 1) directly use Chinese words as vocab (*seg*) does not perform well; 2) *seg_tok* based ALBERT consistently performs better than *char* and *seg* on sentence classification tasks, but not sequence labeling tasks; 3) MVP strategies can help to improve a single vocab model on both types of tasks, especially it can help *seg_tok* on sequence labeling tasks; 4) MVP$_{pair}$ ensemble are the best model, but it comes with a higher inference time; 5) MVP$_{obj}$ help to provide a better single vocab model than MVP$_{pair}$.

We now summaries the following contributions in this work.

- experiments on three ways of building new vocab for Chinese BERT, and *seg_tok*, the combination of CWS and subword tokenization is novel.
- We propose 3 MVP strategies for enhancing the Chinese PLMs.

## 2 RELATED WORK

Before and since Devlin et al. (2018), a large amount of literature on pretrained language model appear and push the NLP community forward with a speed that has never been witnessed before. Peters et al. (2018) is one of the earliest PLMs that learns contextualized representations of words. GPTs Radford et al. (2018; 2019) and BERT Devlin et al. (2018) take advantages of Transformer Vaswani et al. (2017). GPTs are uni-directional and make prediction on the input text in an auto-regressive manner, and BERT is bi-directional and make prediction on the whole or part of the input text. In its core, what makes BERT so powerful are the pretraing tasks, i.e., Mask language modeling (MLM) and next sentence prediction (NSP), where the former is more important. Since

BERT, a series of improvements have been proposed. The first branch of literature improves the model architecture of BERT. ALBERT Lan et al. (2019) makes BERT more light-weighted by embedding factorization and progressive cross layer parameter sharing. Zaheer et al. (2020) improve BERT's performance on longer sequences by employing sparser attention.

The second branch of literature improve the training of BERT. Liu et al. (2019b) stabilize and improve the training of BERT with larger corpus. More work have focused on new language pretraining tasks. ALBERT Lan et al. (2019) introduce sentence order prediction (SOP). StructBERT Wang et al. (2019) designs two novel pre-training tasks, word structural task and sentence structural task, for learning of better representations of tokens and sentences. ERNIE 2.0 Sun et al. (2019) proposes a series of pretraining tasks and applies continual learning to incorporate these tasks. ELEC-TRA Clark et al. (2020) has a GAN-style pretraining task for efciently utilizing all tokens in pre-training. Our work is closely related to this branch of literature by design a series of novel pretraining objective by incorporating multiple vocabularies. Our proposed tasks focus on intra-sentence contextual learning, and it can be easily incorporated with other sentence structural tasks like SOP.

Another branch of literature look into the role of words in pre-training. Although not mentioned in Devlin et al. (2018), the authors propose whole word masking in their open-source repository, which is effective for pretraining BERT. In SpanBERT Joshi et al. (2019), text spans are masked in pre-training and the learned model can substantially enhance the performances of span selection tasks. It is indicated that word segmentation is especially important for Chinese PLMs. Cui et al. (2019a) and Sun et al. (2019) both show that masking tokens in the units of natural Chinese words instead of single Chinese characters can significantly improve Chinese PLMs. In this work, compared to literature, we propose to re-design the vocabulary of the Chinese BERT by combining word segmentation and sub-word tokenizations.

## 3 OUR METHODS

In this section, we present our methods for rebuilding the vocab for Chinese PLMs, and introduce our series of MVP strategies.

### 3.1 BUILDING THE VOCABS

We investigate four work-flows to process the text inputs, each corresponding to a different vocab (or a group of vocabs) (Figure 1). We first introduce the single vocab models, *char*, *seg_tok* and *seg*. For char based vocab *char*, Chinese characters in the corpus are treated as words in English and are separated with blank spaces and a sub-word tokenizer is learned. *seg_tok* requires the sentences in corpus to be segmented and a sub-word tokenizer like BPE are learned on the segmented sentences. Note that in *seg_tok*, some natural word will be splitted into pieces, but there are still many tokens that have multiple Chinese chars. Finally, *seg* with size $N$ is built with the following procedures: a) do CWS on the corpus; b) count the words' frequency, and add the tokens from *seg_tok* with frequency 0; c) for long tail Chinese words and non-Chinese tokens, tokenize them into subword tokens with *seg_tok*, and the words' frequencies are added to the sub-word tokens' frequencies; d) sort the vocab via frequency, and if the most frequent N words or tokens can cover 99.95% of the corpus, then take them as vocab. Note that some of the tokens from *seg_tok* will be dropped.

### 3.2 MULTI-VOCAB PRETRAINING (MVP)

In this subsection, we will introduce MVP, which is a derivation of MLM task by Devlin et al. (2018). MVP has three versions, $\text{MVP}_{hier}$, $\text{MVP}_{obj}$ and $\text{MVP}_{pair}$, all of which have one thing in common, that is, they require more than one vocab to implement pre-training.

Figure 2(a) depicts the architecture of $\text{MVP}_{hier}$ and Figure 2(a) depicts its procedure for processing input sentences. Two vocab, a fine-grained vocab $V_f$, and a more coarse-grained vocab $V_c$, are combined in a hierarchical way. Sequences are first tokenized via $V_c$, and then the Chinese tokens (if containing multiple Chinese characters) are splitted into single characters. Chinese characters and non-Chinese tokens are embedded into vectors. Then representations of chars inside a word is aggregated into the representation of this word, which is further fed into the transformer encoder. During MLM task, whole word masking is applied, that is, we will mask 15% of the tokens

in the $V_c$. For example in Figure 2(a), "喜欢" (like) is masked, thus in the char sequence, two tokens "" $and$ "" $are masked. Then an aggregator will combine the embeddings into the vectors of word tokens. At the MLM task, a$ Let $\mathbf{x}$ and $\mathbf{y}$ denote the sequences of tokens with length $l_x$ and $l_y$, for the same sentence under $V_c$ and $V_f$, in which a part of tokens are masked. Denote $\mathbf{x}^{mask}$ as the masked tokens under $V_c$. The loss function for MVP$_{hier}$ is

$$\min_\theta -\log P_\theta(\mathbf{x}^{mask}|\mathbf{x}, \mathbf{y}) \approx \min_\theta -\sum_{i=1}^{l_x} I_i \log P_\theta(x_i^{mask}|\mathbf{x}, \mathbf{y}),$$

in which $I_i^x$ is a variable with binary values indicating whether the $i$-th token is masked in $\mathbf{x}$.

In MVP$_{obj}$, a sentence is tokenized and embedded in a fine-grained $V_f$ (e.g., a char based vocab), and MLM task on $V_f$ is conducted. However, different from the vanilla MLM, another MLM task based on a more coarse-grained vocab $V_c$ is added. For example, encoded representations of the chars "喜" and "欢" inside the word "喜欢" is aggregated to the vector representation of the word, and an auxiliary MLM layer is tasked to predict the word based on $V_c$. For the aggregator in the example, we adopt BERT-style pooler, which is to use the starting token's representation to represent the word's representation.[4] Denote $\mathbf{x}^{mask}$ and $\mathbf{y}^{mask}$ as the masked tokens under $V_f$ and $V_c$ respectively. The loss function for MVP$_{obj}$ is as follows:

$$\min_\theta -\log P_\theta(\mathbf{x}^{mask}, \mathbf{y}^{mask}|\mathbf{x}) \approx \min_\theta -\sum_{i=1}^{l_x} I_i^x \log P_\theta(x_i^{mask}|\mathbf{x}) - \lambda * \sum_{i=1}^{l_y} I_i^y \log P_\theta(y_i^{mask}|\mathbf{x}),$$

in which $I_i^x$ and $I_i^y$ are variables with binary values indicating whether the $i$-th token is masked in sequence x and y respectively. Here $\lambda$ is the coefficient which measures the relative importance of the auxiliary MLM task.

Now we introduce MVP$_{pair}$, which is the most resource-demanding version of MVP, but it will be proven beneficial. Our goal is to enhance the model with a single vocab, by introducing additional encoder with another vocab and a corresponding MLM task. In this strategy, a sentence is tokenized and embedded with two vocabs, $V_1$ and $V_2$, which will be fed into separate transformer encoders. Transformer encoders will share the same parameters, but the embedding layers are separate. For example in Figure 2(c), "喜欢" is masked, and thus two tokens on the left sequence and(or) one token on the right sequence are masked. The left encoder and MLM layer is tasked to recover the two tokens "喜" and "欢". And on the right, the other MLM layer needs to recover "喜欢". Through parameter sharing, self-supervised signals from one vocab is transferred to the model with the other vocab. Formally, Thus the loss function for MVP$_{pair}$ is

$$\min_\theta -\log P_\theta(\mathbf{x}^{mask}, \mathbf{y}^{mask}|\mathbf{x}, \mathbf{y}) \approx \min_\theta -\sum_{i=1}^{l_x} I_i^x \log P_\theta(x_i^{mask}|\mathbf{x}) - \lambda * \sum_{i=1}^{l_y} I_i^y \log P_\theta(y_i^{mask}|\mathbf{y}),$$

If after MVP$_{pair}$ pretraining we decide to only keep one of the encoder, we will call this model single vocab MVP$_{pair}$. Otherwise, we can call the model as ensemble MVP$_{pair}$. For single mode MVP$_{pair}$, finetuning is the same with vanilla ALBERT. Finetuning for ensemble mode MVP$_{pair}$ is different. For sentence classification, the pooled vectors on both [CLS] tokens will be concatenated to be the feature vector of the classifier. When doing sequence labeling tasks, two ways of ensemble can be conducted, which we will use the example from Figure 2(c) to illustrate. The first approach is to concatenate the features from fine-grained encoder to the coarse-grained encoder. That is, representations of "喜" and "欢" are aggregated to the representation of "喜欢", and it will be concatenated to the representation from the coarse-grained encoder on the right. We will call this approach fine-to-coarse ensemble. The other approach is coarse-to-fine ensemble, which is to concatenate the representation of "喜欢" from the coarse-grained encoder to "喜" and "欢" from the fine-grained encoder. Then the labels of "喜" and "欢" are predicted.

For notational convenience, we will denote the model pretrained with MVP$_{hier}$ strategy and vocab $V$ as MVP$_{hier}(V)$. MVP$_{obj}$ with a fine-grained vocab $V_f$ and a coarse-grained vocab $V_c$ are denoted

---

[4]Due to limited resources available, we leave to future work to investigate whether alternative aggregators can bring improvements.

as $\text{MVP}_{obj}(V_f, V_c)$. $\text{MVP}_{pair}$ with two vocab $V_1$ and $V_2$ is denoted as $\text{MVP}_{pair}(V_1, V_2, i, j)$, where $i$ can be $single$, meaning only keep the encoder from $V_1$ for finetuning, or $ensemble$, meaning keep both encoders, and $j$ can be $ftc$ (short for fine-to-coarse) or $ctf$ (coarse-to-fine), which is two approaches for finetuning $\text{MVP}_{pair}$ on sequence labeling tasks. Note that if $i$ equals $single$, we will neglect the parameter $j$.

## 4 EXPERIMENTS

### 4.1 SETUP

For pre-training corpus, we use Chinese Wikipedia. The vocab size is 21128 for *char* and *seg_tok*. 50.38% of *seg* are Chinese tokens with length more than 1.

In this article, we adopt jieba as our CWS tool.[5] To keep 70% of the vocab as natural words from CWS, and cover 99.95% of the corpus, *seg* is built with vocab size 69341.

For $\text{MVP}_{hier}$, we consider $\text{MVP}_{hier}(seg\_tok)$. For $\text{MVP}_{obj}$, we consider $\text{MVP}_{obj}(char, seg\_tok)$ and $\text{MVP}_{obj}(seg\_tok, seg)$, with $\lambda$ equal to 0.1, 0.5, 1.0, 2.0, 10.0. For $\text{MVP}_{pair}$, we consider $\text{MVP}_{pair}(char, seg\_tok)$, the combination of *seg_tok* and char based vocab, with $\lambda$ equal to 0.1, 0.5, 1.0, 2.0, 10.0. Note that to maintain the consistency for non-Chinese tokens, the char based vocab in $\text{MVP}_{hier}(seg\_tok)$, $\text{MVP}_{obj}(char, seg\_tok)$ and $\text{MVP}_{pair}(char, seg\_tok)$ is derived from *seg_tok* by splitting Chinese characters of *seg_tok* tokens into characters, not exactly the same with *char*.

For pretraining, whole word masking is adopted, and total 15% of the words (from CWS) in the corpus are masked, which are then tokenized into different tokens under different vocabs. For $\text{MVP}_{obj}(char, seg\_tok)$ and $\text{MVP}_{pair}(char, seg\_tok)$, 1/3 of the time only tokens *seg_tok* are predicted, and 1/3 of the time only tokens from the derived char based vocab are predicted, and for the rest of the time, tokens from both vocabs are predicted.

In this article, all models use the ALBERT as encoder. We make use of a smaller parameter settings, that is, the number of layer is 3, the embedding size is 128 and the hidden size is 256. Other ALBERT configurations remain the same with ALBERT Lan et al. (2019). The pretraining hyper-parameters are almost the same with ALBERT Lan et al. (2019). The maximum sequence length is 512. Here, the sequence length is counted under the coarse-grained vocab for $\text{MVP}_{hier}$, fine-grained vocab for $\text{MVP}_{obj}$, and the longer one of the two sequences under the two vocabs for $\text{MVP}_{pair}$. The batch size is 1024, and all the model are trained for 12.5k steps. The pretraining optimizer is LAMB and the learning rate is 1e-4. For finetuning, the sequence length is 256, the learning rate is 2e-5, the optimizer is Adam Kingma & Ba (2015) and the batch size is set as the power of 2 and so that each epoch contains less than 1000 steps. Each model is run on a given task for 10 times and the average performance scores and standard deviations are reported for reproducibility.

### 4.2 BENCHMARK TASKS

For downstream tasks, we select 4 text classification tasks: (1) ChnSentiCorp (chn)[6], a hotel review dataset; (2) Book review[7] (book_review), collected from Douban[8] by Liu et al. (2019a). For sentence pair classification tasks, we include the following 4 datasets: (1) XNLI (xnli) from Conneau et al. (2018) ; (2) LCQMC (lcqmc) Liu et al. (2018); (3) NLPCC-DBQA[9] (nlpcc_dbqa), a question-answer matching task in the open domain; (4) Law QA Liu et al. (2019a), a QA matching task in the legal domain. We also investigate three NER tasks. MSRA NER (msra) Levow (2006) is from open

---

[5]Despite there are more sophisticated CWS toolkits avalable, and using them may lead to better performances, jieba is efficient and good enough to prove the importance of word segmentation in the vocab design of Chinese BERT.

[6]https://github.com/pengming617/bert classification

[7]https://embedding.github.io/evaluation/

[8]https://book.douban.com/

[9]http://tcci.ccf.org.cn/conference/2016/dldoc/evagline2.pdf

| task | chn | br | lcqmc | xnli | nlpcc | msra | fin | ccks |
|---|---|---|---|---|---|---|---|---|
| metric | acc | acc | acc | acc | macro F1 | exact F1 | exact F1 | exact F1 |
| char | 86.61 ± 1.12 | 77.83 ± 0.55 | 77.85 ± 0.78 | 59.22 ± 0.76 | 62.09 ± 2.57 | **81.14** ± 0.61 | **73.43** ± 0.61 | **85.63** ± 0.28 |
| seg_tok | **87.17** ± 0.46 | **79.08** ± 0.20 | **79.79** ± 0.42 | **60.19** ± 0.43 | **64.02** ± 1.15 | 79.81 ± 0.81 | 72.12 ± 1.26 | 84.79 ± 0.74 |
| seg | 87.06 ± 0.78 | 78.53 ± 0.43 | 79.27 ± 0.68 | 59.71 ± 0.59 | 63.32 ± 1.83 | 79.07 ± 1.17 | 71.24 ± 1.64 | 83.96 ± 0.93 |
| $\text{MVP}_{hier}(seg\_tok)$ | 87.06 ± 0.48 | 78.67 ± 0.53 | 79.64 ± 0.59 | 59.89 ± 0.53 | 64.06 ± 1.32 | 80.57 ± 0.75 | 72.36 ± 1.07 | 84.88 ± 0.98 |
| $\text{MVP}_{obj}(char, seg\_tok)$ | 87.26 ± 0.52 | 78.45 ± 0.64 | 78.92 ± 0.56 | 59.98 ± 0.48 | 63.67 ± 1.22 | 81.56 ± 0.56 | 73.95 ± 0.58 | 86.10 ± 0.35 |
| $\text{MVP}_{obj}(seg\_tok, seg)$ | 87.55 ± 0.32 | 79.67 ± 0.21 | 80.44 ± 0.39 | 60.57 ± 0.36 | 65.48 ± 0.77 | 81.05 ± 1.23 | 73.47 ± 0.68 | 85.56 |
| $\text{MVP}_{pair}(char, seg\_tok, s)$ | 87.04 ± 0.66 | 78.56 ± 0.68 | 78.63 ± 0.63 | 60.23 ± 0.47 | 63.78 ± 1.28 | 81.47 ± 0.43 | 73.78 ± 0.78 | 85.97 ± 0.45 |
| $\text{MVP}_{pair}(seg\_tok, char, s)$ | 87.22 ± 0.41 | 79.43 ± 0.31 | 80.05 ± 0.44 | 60.25 ± 0.52 | 64.79 ± 1.03 | 80.03 ± 0.85 | 73.17 ± 1.18 | 85.45 ± 0.47 |
| $\text{MVP}_{pair}(seg\_tok, char, e, ftc)$ | **87.96** ± 0.37 | **80.05** ± 0.18 | **80.56** ± 0.32 | **60.86** ± 0.29 | **65.94** ± 0.94 | 81.89 ± 0.65 | 73.97 ± 0.56 | 86.27 ± 0.36 |
| $\text{MVP}_{pair}(char, seg\_tok, e, ctf)$ | - | - | - | - | - | **82.28** ± 0.54 | **74.32** ± 0.47 | **87.85** ± 0.32 |

Table 1: Main results on the Chinese benchmark datasets. For each task and each model, experiments are repeated for 10 times, and the average and standard deviation of the scores are reported.

domain, Finance NER [10] (fin) is from financial domain, and CCKS NER[11] (ccks) is collected from medical records.

### 4.3 EXPERIMENTAL RESULTS

We first compare the three single vocab models. We find that word based vocabs *seg_tok* and *seg* perform better than *char* on all sentence classification tasks, and *seg_tok* outperforms *seg* even though having less parameters. However, *seg_tok* is worse than or comparable to *char* on three NER tasks, where these performance gaps are only partially explained by the CWS errors[12]. We believe that its disadvantages on sequence labeling tasks is due to sparcity of tokens, i.e., each token has less training samples, thus are less well trained for token level classification. From the above results, we show that both CWS and subword tokenization are important to improve the expressiveness of Chinese PLMs in sentence level tasks via vocab re-designing. In addition, *seg_tok* is more efficient than *seg_tok*. we can observe a 1.35x speed up when changing the vocab from *char* to *seg_tok*.

The lower rows of Table 1 also demonstrates the effectiveness of MVP training. Note that in this table, we report $\text{MVP}_{obj}$ with $\lambda = 2.0$, and $\text{MVP}_{pair}$ with $\lambda = 1.0$. First, we can see MVPs consistently improve the models' performances, among which the largest improvements come from $\text{MVP}_{obj}$ and ensemble $\text{MVP}_{pair}$. Second, for *seg_tok*, keeping only one encoder provides improvements on sequence labeling tasks, but the improvements are less than that provided by $\text{MVP}_{pair}(seg\_tok, seg)$ on sentence level tasks. Third, note that for NER tasks, ensemble $\text{MVP}_{pair}$ performs better than single $\text{MVP}_{pair}$, and coarse-to-fine ensemble is more effective than fine-to-coarse ensemble. Fourth, $\text{MVP}_{obj}(seg\_tok, seg)$ achieve performances that are close to $\text{MVP}_{pair}(seg\_tok, char, e)$ on lcqmc, with less than half of the time in pre-training and fine-tuning. But the model's improvements on NER tasks are less significant. Notice that for pretraining and finetuning, $\text{MVP}_{pair}$ takes almost twice the resources and time than $\text{MVP}_{obj}$. Thus, for resources limited scenarios, $\text{MVP}_{obj}$ could be a more suitable choice.

### 4.4 EFFECTS OF $\lambda$

In this section, we investigate how the relative importance coefficient $\lambda$ for $\text{MVP}_{pair}(char, seg\_tok)$ affects the ALBERT models' downstream performances. Here for $\text{MVP}_{pair}$, $\lambda$ is assigned to the loss term from coarse-grained vocab. We set $\lambda$ equal to 0.1, 0.5, 1, 2, 10, and run per-

---

[10]https://embedding.github.io/evaluation/extrinsic

[11]https://biendata.com/competition/CCKS2017_2/

[12]CWS errors accounts for only less than 0.3% of the errors on CCKS.

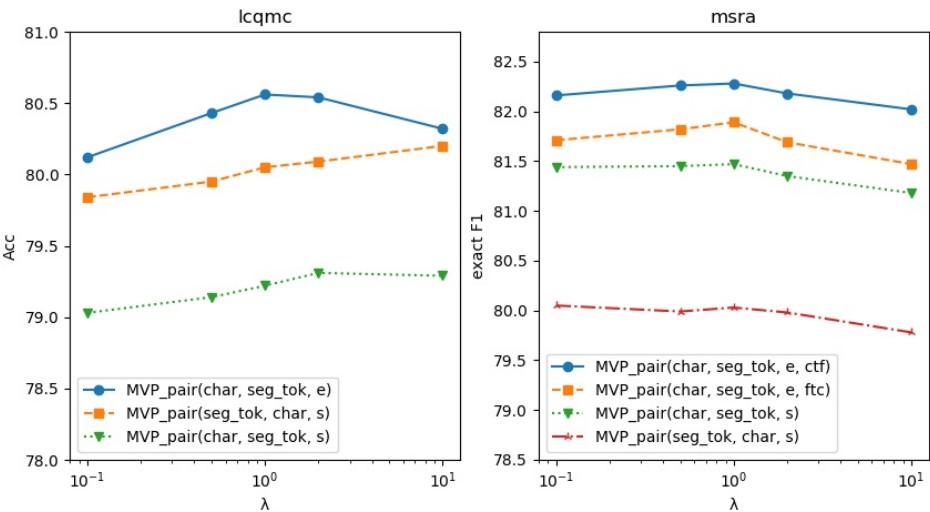

Figure 3: How the value of coefficient $\lambda$ affects the models' downstream performances.

training under each of the setting. Finetuning results on *lcqmc* and *msra* are reported in Figure 3. For ensemble $\text{MVP}_{pair}(char, seg\_tok, e)$, $\lambda = 1.0$ provides the best performances. And for $\text{MVP}_{pair}(char, seg\_tok, s)$, increasing $\lambda$ worsen the performance on NER, but not on sentence level tasks, and vice versa for $\text{MVP}_{pair}(seg\_tok, char, s)$.

## 5 CONCLUSIONS

In this work, we first propose a novel method, *seg_tok*, to re-build the vocabulary of Chinese BERT, with the help of Chinese word segmentation (CWS) and subword tokenization. Then we propose three versions of multi-vocabulary pretraining (MVP) to improve the models' performance. Experiments show that: (a) compared with char based vocabulary, *seg_tok* does not only improves the performances of Chinese PLMs on sentence level tasks, it can also improve efficiency; (b) MVP improves PLMs' downstream performance, especially it can improve *seg_tok*'s performances on sequence labeling tasks.

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
