# OpenReview forum: "MVP-BERT: Redesigning Vocabularies for Chinese BERT and Multi-Vocab Pretraining"
_ICLR.cc/2021/Conference — Reject_

### Official Review · AnonReviewer1 · 2020-10-19
**Not well organised, comparison is not enough**

**Rating:** 3
**Confidence:** 4

**Review:**

This paper compared three different Chinese language pretraining methods. The paper is not well organized. Although the paper mentioned the char and word are both important in the representation of the Chinese language, it is not clear why the author(s) used the current three pretraining methods.  More intuitive explanation of the design of the pretraining structure should be added. The most intuitive way of combining the pretrained word and character information is to pretrain them separately and concatenate them together, the proposed models should compare with this most intuitive method and also explain why the proposed models are better than this simply pretraining concatenation method.

The experiments were not persuasive. Although it is not necessary to beat all the state-of-the-art models, the comparison with other models should be given. For example, the MSRA NER in this paper is only 82% which is largely behind the SOTA models (>93%).  Given the poor baseline performance, it is hard to persuade the readers the conclusions in this paper are still hold in the most recent/advanced models.

Page 2, citation error "Albert ?"
Page 5, format error, the sentence is over the page boundary.

---

### Official Review · AnonReviewer2 · 2020-10-21
**Too poor novelty and writing**

**Rating:** 2
**Confidence:** 5

**Review:**

This paper proposes to use multiple levels of language units, including characters, subwords, and words, for Chinese language modeling. Three versions of multi-vocabulary pretraining methods are also studied. Experiments show that using the optimized seg_tok units could improve the model performance in various downstream tasks, and the MVP strategies boost the seg_tok’s results on sequence labeling tasks.

Strengths:
1.	Empirical studies on building vocabularies with three different granularities of language units for Chinese language model pre-training.
2.	Evaluations on three multi-vocab pre-training strategies.

My most concerns are:
1.	The idea of modeling different levels of language units has been widely studied before. Although they have not been comprehensively evaluated on pre-trained LMs, the findings are quite similar with previous studies, without much new highlights. The claim of the contribution, “the combination of CWS and subword tokenization is novel.”, is quite weak.

Missing References:
Zhang, X., & Li, H. (2020). AMBERT: A Pre-trained Language Model with Multi-Grained Tokenization. arXiv preprint arXiv:2008.11869.
Zhang et al, (2019). "Effective subword segmentation for text comprehension." IEEE/ACM Transactions on Audio, Speech, and Language Processing 27.11 (2019): 1664-1674.
Liu, Z., Xu, Y., Winata, G. I., & Fung, P. (2019). Incorporating word and subword units in unsupervised machine translation using language model rescoring. arXiv preprint arXiv:1908.05925.

2. In seg vocab development, the choice of 99.95% is interesting but can you please tell what is the lowest word frequency in this vocab? how many words with a frequency of less than 40 are present? if criteria is changed from 99.95% coverage to the selection of only frequent words (maybe >40), would it had any effect on speed? or performance?

3.	This paper is poorly written. Thorough proofreading is required. There are too many typos and grammar errors. e.g.,
In the abstract, performances -> performance; remain -> remains; does not only improves -> does not only improve;
In the second paragraph of page 2, incorporate -> incorporates; combine -> combinesl
In Footnote 5, avalable -> available.

4.	There is no comparison on public tasks or datasets.

Other comments:
1.	For the multi-vocabulary pre-training, the vocabulary size, i.e., the size of the seg_toks, would be an important influence factor for the downstream task performance.
2.	Only ALBERT small model is evaluated. It is not quite sure if the method can further enhance the state-of-the-art language models.
3.	The citation format is in chaos, check the usage of \citep{} and \citet{}. The citation of ALBERT is missed in the last paragraph of page 2.
4.	Check the second line in page 5.

---

### Official Review · AnonReviewer4 · 2020-10-27
**Interesting ideas and consistent but marginal improvements over baselines. Miss some related works on this topic.**

**Rating:** 5
**Confidence:** 4

**Review:**

The paper focuses on pretrain techniques for Chinese vocabulary. The paper proposes several interesting and novel ways to improve the pretrained model in Chinese. These ideas include new ways of combining character, segment and word level tokens and new MLM tasks with different granularity of Chinese vocab. For example MVP_hier explicitly combines the chars into words, which doesn't give the best results. MPV_obj outperforms MPV_hier by teaching the transformer to learn how to aggregate meaning of chars with an additional loss with a coarse-grained vocab. The results show that MVP pretrain can improve the performance on both sentence level tasks and sequence labeling tasks. Improvements are consistent over all testing tasks although the improvements appear to be marginal, especially in comparison to the reported standard deviation. For example, MVP_obj outperforms the best baseline fin by 0.52 while the std is 0.58 for MVP_obj and 0.61 for the best baseline. The best results are given by MVP_pair ensemble methods. Usually, ensemble methods can boost the performance of many ML tasks.

It is a bit surprising that the seg_tok is worse than char on sequence labeling tasks. It will be useful to give some intuition or examples to illustrate this. The author mentioned that the CWS attributes very little to the performance loss. It will be helpful if more details are given. The whole technique proposed by the paper also heavily relies on the performance of CWS.

Because the paper proposes a novel technique to work with Chinese characters, one more thing that the paper lacks is to mention and compare with recent works on Chinese language model pretrain and techniques to deal with Chinese vocab. The baseline method is vanilla MLM which was developed for English.

One editing issue: page 5 the second line.

---

### Official Review · AnonReviewer3 · 2020-10-27
**Overall experiments are solid and convincing, but the methods are somewhat lack of novelty.**

**Rating:** 4
**Confidence:** 3

**Review:**

    This work attempt to handle the vocabulary problem in Chinese pre-training, which is indeed an unsolved problem (for comparison, byte-pair encoding has been dominant in English pre-training models). Recently, there are some work trying to combine char-based vocab and word-based vocab for Chinese pre-training, such as AMBERT (Zhang et al) and CharBERT (Ma et al). Compared with these works, this paper is somewhat incremental and lacks novelty. In particular, this paper proposed the multi-vocab pretraining (MVP) and designed three versions of MVP: MVP_hier, MVP_obj, and MVP_pair (which is similar to AMBERT). The authors conducted solid experiments for multiple levels of vocabularies and the several MVP variants on eight Chinese benchmark datasets, which induced some useful take-away findings. Overall, the paper focused on the problem of vocabulary in Chinese pre-training, and experimented several MVP methods. Considering the contributions of this paper, it is more suitable as a technical report than a paper to be published.

    Pros:

    1. The overall experiments are solid and convincing. The implementation details including resources and libraries are sufficient, the experiments are repeated for multiple times for reproducibility.
    2. To handle the problem of Chinese vocabulary, this paper has tested three single-vocab models (char, seg, seg_tok) and several multi-vocab models (MVP). The results and findings demonstrated in this paper should be trusted and also instructive to peers who is interested in the Chinese pre-training.

    Cons:

    1. The methods are somewhat lack of novelty. Few insightful conclusions are revealed either theoretically or experimentally.
    2. The effect of \lambda is not a good point to be analyzed. Instead, there are several interesting points that are more worth to be studied such as the interaction/compositional relationships between multiple vocabs.
    3. The writing should be improved. Too many errors in the current version of paper. For example:
       1. The form of citation should be corrected, e.g., "*The pretrained language models (PLMs) including BERT Devlin et al. (2018)*" should be "*The pretrained language models (PLMs) including BERT (Devlin et al., 2018)*"
       2. In Page 2, the last sentence of the paragraph above Figure 1, "We will denote this strategy as MVP_pair", I think the "MVP_pair" should be "MVP_obj" .
       3. The second row in page 5.
       4. seg_tok is not a good name. The authors should take more time to completing this paper.

---

### Decision · Program_Chairs · 2021-01-07
**Final Decision**

**Decision:**

Reject

**Comment:**

Three reviewers agreed to reject and the other reviewer also suggested it is below the threshold.